# Benefits and Harms of Antibiotic Use in End-of-Life Patients: Retrospective Study in Palliative Care

**DOI:** 10.3390/antibiotics14080782

**Published:** 2025-08-01

**Authors:** Rita Faustino Silva, Joana Brandão Silva, António Pereira Neves, Daniel Canelas, João Rocha Neves, José Paulo Andrade, Marília Dourado, Hugo Ribeiro

**Affiliations:** 1Faculty of Medicine, University of Coimbra, Azinhaga de Santa Comba, Celas, 3000-548 Coimbra, Portugal; 2Abel Salazar Institute of Biomedical Sciences, 4050-313 Porto, Portugal; 3RISE-Health, 4200-319 Porto, Portugal; 4Faculty of Medicine, University of Porto, 4200-319 Porto, Portugal; 5Department of Vascular Surgery, Local Health Unit São João, 4200-319 Porto, Portugal; 6Department of Vascular Surgery, Unidade Local de Saúde—Alto Ave, 4835-044 Guimarães, Portugal; 7Coimbra Institute for Clinical and Biomedical Research, 3000-548 Coimbra, Portugal; 8Community Palliative Care Team Gaia, Local Health Unit Gaia and Espinho, 4400-129 Vila Nova de Gaia, Portugal

**Keywords:** terminal care, anti-bacterial agents, signs and symptoms, quality of life, home care services

## Abstract

**Context:** Many patients at the end of life receive antibiotics to alleviate symptoms and improve quality of life; however, clear guidelines supporting decision making about the use of antibiotics are still lacking. **Objectives:** This study aimed to evaluate the benefits and harms of antibiotic use among patients under a palliative care community support team in Portugal. **Methods:** An observational, cross-sectional, retrospective study was conducted on 249 patients who died over a two-year period, having been followed for at least 30 days prior to their death. Data included patient demographics, clinical diagnoses, antibiotic prescriptions, and symptomatic outcomes. The effects of commonly prescribed antibiotics—amoxicillin + clavulanic acid, cefixime, ciprofloxacin, and levofloxacin—were compared using statistical analyses to assess survival, symptom intensity, and functional scales. **Results:** Adverse events, primarily infections and secretions, occurred in 57.8% of cases, with 33.7% receiving antibiotics. No significant difference in survival was observed across the antibiotic groups (*p* = 0.990). Symptom intensity significantly reduced after 72 h of treatment (*p* < 0.05), with ciprofloxacin demonstrating the greatest symptom control. The Palliative Outcome Scale decreased uniformly, with higher scores associated with amoxicillin + clavulanic acid (*p* = 0.004). The Palliative Performance Scale declined post-treatment, with significant changes noted for cefixime and ciprofloxacin (*p* < 0.05). **Conclusions:** Antibiotics may improve symptom control and quality of life in the end-of-life stage. While second-line antibiotics may offer additional benefits, the heterogeneity of the sample and limited adverse effect data underscore the need for further research to guide appropriate prescription practices in palliative care.

## 1. Introduction

Specialized palliative care (PC) is a multidisciplinary approach applicable to patients with high clinical complexity. Its main objective is to mitigate multidimensional suffering and improve the quality of life without shortening or prolonging it unnecessarily [1].

Several clinical signs suggest that an individual with advanced disease is living his or her final days; for example, delirium, Cheyne–Stokes breathing pattern, rales, decreased level of consciousness, reduced social interaction, and oligoanuria, among other signs, should alert health professionals and motivate new approaches [2,3].

It is essential to review the therapeutic table and consider which treatments are aligned with the objectives. Invasive methods and medications for controlling the underlying pathology that do not contribute to symptomatic control lose interest, especially when the adverse effects outweigh their benefits [2,3,4,5,6,7,8,9,10,11].

Infections are common complications at the end of life and are often the cause of death. The most common are urinary tract infections and respiratory infections, and the diagnosis is usually clinical [12,13,14]. Antibiotics may be considered even when the patient does not meet the diagnostic criteria for bacterial infection to relieve suggestive symptoms, such as fever or respiratory secretions [12,15]. However, the effectiveness of antibiotics in reducing symptom burden is unpredictable, with documented improvement in 60–92% of urinary tract infections and 0–53% of respiratory infections, but no significant benefits in bacteremia [12]. Consequently, this practice presents a complex ethical and clinical dilemma.

Marra et al. [16] concluded, in a systematic review with meta-analysis, that more than half of patients are exposed to antibiotics during the end-of-life period, and highlighted the fact that existing studies, although informative on the prevalence of antibiotic use in this population, did not address the individual factors that motivated prescription or the effects of their use [16].

The choice of the antibiotic, considered on a case-by-case basis and based on the physician’s experience, can be a challenge. On the one hand, although safer, antibiotics used as first-line therapy in other contexts [17,18,19,20] may not be sufficient in these patients. The benefits to quality of life when opting for second-line therapy may not justify the losses, exposing the patient to more serious adverse effects [21] or requiring hospitalization, which may not be compatible with their wishes [22] and entails higher clinical and organizational costs [23]. In addition, the selection of multidrug-resistant microorganisms resulting from antibiotic therapy with broad-spectrum drugs poses a danger to public health. It puts the patient at risk of developing new, potentially more serious infections, forcing hospitalization and isolation, reducing the quality of life, or even shortening the patient’s life [24].

A retrospective study conducted at a palliative care center in Sweden sought to clarify the benefit of antibiotics in treating end-of-life cancer patients [25]. It was found that 37% of patients who received antibiotics experienced significant symptom relief, suggesting that their use may benefit patients, particularly those with sepsis; however, a separate retrospective study conducted in 2021 suggested the contrary [26]. It considered 133 patients admitted to a palliative care unit in the last 14 days of life, diagnosed with infection, of whom 90 received antibiotics. It found that during this period, there were no significant differences in the frequency of documented symptoms between patients who received antibiotic therapy and those who did not, suggesting no benefit to their use. However, considering only the presence or absence of symptoms, it did not discriminate differences in intensity and their impact on quality of life and functionality [26].

Consequently, despite the widespread use of antibiotics in end-of-life care, there is a significant lack of robust evidence to guide appropriate prescription practices. Existing guidelines are often vague, and there is a dearth of data on antibiotic-specific outcomes in different palliative care settings and populations. In particular, there is a paucity of data on the benefits and harms of specific antibiotics in Portuguese palliative care.

To address this research gap, this retrospective study aimed to evaluate the effectiveness and safety of commonly prescribed antibiotics in a cohort of patients receiving palliative care in the community in Northern Portugal. Our objectives were to (1) assess the impact of antibiotic use on symptom intensity, quality of life, functionality, and survival; (2) compare the outcomes associated with different antibiotics; and (3) identify potential risk factors for adverse events.

## 2. Results

The study included a total of 249 patients. The average age at the start of follow-up by the team was 81.5 years, ranging from a minimum of 37 to a maximum of 108 years. A total of 135 patients were female (54.2%) and 114 were male (45.8%). The most frequent prognostic-defining disease was cancer (140 patients; 56%), followed by frailty syndrome (45 patients; 18%), dementia (42 patients; 17%), and heart failure (22 patients; 9%). The average follow-up period was 29.6 days, with a median of 14 days.

Of the 249 patients, antibiotics were administered to 66 (26.5%), motivated by clinical conditions such as skin infections, urinary tract infections, pneumonia, and respiratory secretions. The clinical conditions observed did not differ based on the pathology most likely to define the prognosis (χ^2^ (12) = 15.297, *p* = 0.222).

It is important to clarify that respiratory secretions were not classified as infections per se, but rather as symptomatic presentations that may benefit from antibiotic therapy for symptom control. The use of antibiotics for respiratory secretion management in palliative care is a well-established practice aimed at reducing secretion viscosity and bacterial overgrowth rather than treating confirmed infection. This approach is recommended in palliative care guidelines for symptom management rather than infection treatment [27].

We observed a significantly higher proportion of antibiotic use in patients with heart failure (17 out of 22 patients; 77.3%) and dementia (32 out of 42 patients; 76.2%) (χ^2^ (3) = 33.487, *p* < 0.001).

There was a difference in follow-up times, with a median of 21.5 days in patients medicated with antibiotics and 10 days in patients not medicated with antibiotics (*p* < 0.001). The age difference between medicated and non-medicated patients was not statistically significant (t (242) = −0.679, *p* = 0.498).

There were no significant differences in patient survival vs. antibiotic used (χ^2^ KW (3) = 0.112, *p* = 0.990).

A statistically significant reduction in symptom intensity was observed in all patients 72 h after treatment with antibiotics (Table 1), regardless of the drug used (*p* < 0.05). Prior to this, there were no significant differences between the groups (χ^2^ (3) = 6.071, *p* = 0.108). However, 72 h after the start of treatment, the group treated with amoxicillin + clavulanic acid showed greater symptom intensity (*n* = 4.25/10) than the group receiving ciprofloxacin (*n* = 1.89/10) (χ^2^ KW (3) = 10.104, *p* = 0.018).

Statistical analyses also revealed no significant correlation between the administered doses of antimuscarinics (ipratropium bromide), systemic corticosteroids (dexamethasone), and/or inhaled corticosteroids (fluticasone or budesonide), opioids (morphine equivalent), and patient outcomes. Although these medications were used by all patients, their presence alone without antibiotics also influenced the study results (Table 1 and Table 2).

We also found a statistically significant improvement in symptom control after 72 h (6.84 vs. 4.52) in the 183 patients (73.5%) who did not take antibiotics (Table 1).

POS (Table 2) uniformly decreased after 72 h, regardless of the antibiotic used (*p* < 0.05). The difference between the groups was not significant pre-treatment (*p* = 0.169); however, 72 h after the start of treatment, POS was significantly higher with amoxicillin + clavulanic acid than with ciprofloxacin (*p* = 0.004), i.e., patients treated with amoxicillin + clavulanic acid had a worse quality of life (POS 1.95 ± 1.05) compared with patients treated with ciprofloxacin (POS 0.94 ± 0.80).

PPS (Table 3) always decreased after treatment, although the differences were only statistically significant with cefixime (Z = −2.070, *p* = 0.038) and ciprofloxacin (Z = 2.8228, *p* = 0.005).

The correlation between POS and the intensity of symptoms before treatment was statistically significant, positive, and moderate (r sp = 0.691, *p* < 0.001). The same was true for POS and intensity of symptoms 72 h after the start of therapy (r sp = 0.458, *p* = 0.042).

Complications were observed in 15 patients (17.9%) treated with the first antibiotic, with the main complication recorded being treatment failure (*n* = 10), followed by new infection (*n* = 4) and relapse (*n* = 1). Of the patients who received a second antibiotic (*n* = 18), one experienced treatment failure and one reported an episode of seizures. The seizures occurred after the initiation of treatment with ciprofloxacin.

## 3. Discussion

Our results suggest that the four most commonly used antibiotics could significantly reduce the intensity of symptoms and contribute to an improvement in the quality of life for all patients with high clinical complexity due to severe disease (oncological and non-oncological) when they have a clinical condition suggestive of community infection.

Although the four most commonly used antibiotics led to improvements, patients who received amoxicillin + clavulanic acid or cefixime experienced more intense symptoms 72 h after treatment initiation compared with those who received the other antibiotics. On the other hand, patients treated with ciprofloxacin presented the fewest symptoms. These results suggest a need to use second-line antibiotics more frequently in fragile/high clinically complex patients. However, it is difficult to draw clear conclusions because of the issues inherent in comparing different individuals with dissimilar clinical conditions.

All patients had concurrent symptomatic medications already established, including opioids, corticosteroids, and anti-secretory medications. No statistically significant differences were observed in dosing changes for these medications between antibiotic treatment periods. These concurrent therapies were maintained throughout the study to ensure consistent baseline symptomatic management.

Other authors observed a tendency to administer more antibiotics to patients with heart failure or dementia, and they thought that this may be associated with the natural course of the diseases, which in these cases allows prolonged survival even with very severe frailty [28]. We found that patients with heart failure or dementia received more antibiotics, although the underlying disease did not appear to be associated with the type of complications recorded. As expected, the follow-up time was longer for patients receiving antibiotic therapy. This potentially reveals the team’s good understanding of life expectancy and knowledge of the expected course of each disease and individual, allowing them to make decisions accordingly.

Other studies have found that treatment with antibiotics is likely more common in younger patients, where there is a greater likelihood of improvement [26]. Our study addresses the potential for age bias in treatment decisions, particularly regarding the use of antibiotics. We found that age was not a determining factor in the CPCT decision-making process, which aligns with its commitment to patient-centered care and individualized treatment.

Patients treated with antibiotics showed a significant improvement in quality of life; in particular, those treated with ciprofloxacin showed a more marked improvement. Given the positive and moderate correlation found between POS and the intensity of symptoms, it makes sense that an antibiotic with superior symptom control would yield better results in terms of quality of life. However, it is important to emphasize that we are considering all clinical situations without particularizing the patient. For example, bronchorrhea caused by the use of an antibiotic, particularly ciprofloxacin, can be more harmful in patients who cannot expel respiratory secretions. It will be interesting to explore this therapeutic personalization in the future, with studies aimed at the diagnosis that motivated the use of antibiotics, measuring outcomes identical to those we presented in our study.

Even when death is expected and the goals of treatment change to focus on comfort care, other studies have shown that more than a third of patients continue to receive antibiotic therapy [29,30]. Crowley et al. [31] question this practice, stating that although effective in controlling some symptoms, it does not guarantee benefits in quality of life. Chih et al. [32] also noted that using antibiotics in the final two days can impair survival and concluded that discontinuation of antibiotic therapy should be considered when signs of the end of life appear. However, our results suggest that antibiotics provide symptomatic relief and improve quality of life, even when the general goals of care have shifted to focus on comfort.

Functionality, assessed using the PPS, worsened despite antibiotic treatment. This worsening is predictable, given most of the patients’ end-of-life phase and the association between the PPS and survival [28]. A decline in functional capacity is expected as the date of death approaches.

Based on the data, none of the drugs stood out in terms of survival; thus, it would be logical to expect that all of them would have a similar impact on functionality. However, the decrease in PPS was only statistically significant in patients treated with cefixime and ciprofloxacin, suggesting that the use of amoxicillin + clavulanic acid or levofloxacin would have a greater advantage in preserving functionality. It is worth noting that patients treated with amoxicillin + clavulanic acid and levofloxacin had, on average, lower pre-treatment PPS. These patients had more residual functionality before treatment; therefore, the worsening observed at 72 h was not as significant as in patients who retained some functional capacity before receiving the antibiotic, making it difficult to draw conclusions based on these data.

In this study, complications were defined as any new infection, treatment failure, relapse, hospitalization, or adverse event requiring medical intervention. The most frequent complication was therapeutic failure, exemplified by the lack of symptomatic improvement after 72 h. Of these complications, the only documented adverse reaction was a case of convulsions after taking ciprofloxacin. Despite this event, ciprofloxacin achieved the best results in terms of symptom control and quality of life. These data support the hypothesis that choosing antibiotics with a broader spectrum of action and greater bactericidal potential may be the best strategy for treating patients in the final stages of life.

It should be noted that since only the events leading to the introduction of a new antibiotic were considered, adverse effects that may have caused discomfort, although not requiring changes in therapy, were not reported. Data on quality of life suggest that the treatment had an overall positive impact, even in the presence of some undesirable events. However, it is important that future studies rigorously investigate the adverse effects and impacts of antibiotics during the end-of-life period to facilitate their implementation as preventive measures or therapeutic strategies.

Finally, it is essential not to forget the risk of developing antibiotic-resistant microorganisms, one of the most important causes of death globally [33]. This consequence may be easily overlooked, particularly when weighed against the immediate benefits accruing to the individual patient [34]. However, the truth is that, although it does not appear to increase mortality, colonization or infection by multidrug-resistant microorganisms has the potential to harm the patient’s quality of life [24,35]. Although there seem to be advantages in choosing second-line antibiotics as a palliative strategy in patients at the end of life, we cannot rule out the possibility that the harms outweigh these benefits. To us, the major problem with multidrug resistance is not directly associated with palliative care and patients at the end of life, but instead with a failure to adopt evidence-based antibiotic-sparing strategies, as described by the WHO [36], at all stages of people’s lives.

### Study Limitations and Main Strengths

A notable strength of this study lies in its approach to evaluating the impact of various antibiotics, considering not only symptom burden but also the quality of life experienced by patients. In addition, we sought to provide a more objective description of treatment effects, using numerical scales that allow us to identify the changes and magnitudes of treatments.

However, some limitations should be mentioned. The retrospective nature of the study does not allow for the easy establishment of causal relationships, and data collection relied on documentation with potential errors and omissions. However, the missing percentage per column was below 1%.

The study is single-centered, which does not allow for the reproducibility of data using the same database; additionally, the sample size is relatively small, making it difficult for us to reach definitive conclusions about each antibiotic used in end-of-life patients followed by specialized palliative care teams. Furthermore, we only found individual statistical significance for four antibiotics; we are aware that many others can be used by specialized palliative care teams.

Despite our efforts to control for the effects of other medications (opioids, corticosteroids, antimuscarinics), residual confounding may still be present. It is possible that the observed effects of antibiotics on symptom relief and quality of life were partially influenced by unmeasured or poorly measured factors. Future studies should consider using more rigorous methods to control for confounding, such as propensity score matching or instrumental variable analysis.

Another limitation is the absence of objective biomarker data (e.g., C-reactive protein, white blood cell count, procalcitonin) to confirm infection and assess treatment response. Obtaining these biomarkers in frail, community-dwelling palliative care patients can be burdensome and potentially cause more harm than good. Furthermore, our primary aim was to improve symptom control and quality of life, making clinical outcomes the primary focus. Future studies could explore the feasibility and value of incorporating biomarker data in this population, balancing the potential benefits with the risks and burdens.

## 4. Materials and Methods

The study had an observational, cross-sectional, retrospective, and non-interventional design.

It was developed in compliance with ethical and legal principles and was approved by the Ethics Committee of the Faculty of Medicine of the University of Coimbra (CEFMUC 129/2024, 20 November 2024). As this is a non-interventional clinical study with a large number of participants, and considering that the vast majority of patients were already deceased, an exemption from informed consent was granted by the Ethics Committee of the Faculty of Medicine of the University of Coimbra, as provided for in No. 6 of art. 19 of Law No. 12/2005, of January 26, amended by Law No. 26/2016, of August 22, and in accordance with the provisions of No. 2 of art. 6 of the Clinical Research Law No. 21/2004, of April 16.

All patients who died during the two-year period (2023 and 2024) of activity of a community palliative care team (CPCT) in Northern Portugal and who had been followed for a minimum of 30 days were selected to constitute the study sample. Based on these inclusion criteria, we selected a total of 249 patients (36,7%) out of 678 patients followed by the CPCT in 2023 and 2024. We excluded patients with less than 30 days of follow-up by the CPCT because we wanted to assess the palliative medical doctors’ practices. This criterion also enabled us to obtain more uniform and appropriate clinical records for the variables we intended to measure and include in the database.

The duration of follow-up was determined based on the amount of time each patient was under the care of the CPCT before death. Follow-up time was calculated from the date of the first appointment until the date of death. The follow-up time varied, depending on disease progression, referral patterns, and patient/family preferences.

Data were collected from individual clinical records and recorded in a password-protected Microsoft Excel^®^ spreadsheet, with random numbers assigned to ensure patient confidentiality. The database will be deleted once the results have been presented using a data sanitization tool.

The following variables were considered: gender, age, duration of care provided by the team, primary diagnosis, other diagnoses, reason for starting treatments aimed at symptomatic control (distinguishing between the 1st treatment and a 2nd treatment), use of antibiotics (distinguishing between the 1st, 2nd, and 3rd antibiotic), days until the introduction of the 2nd antibiotic, and date of death. Two time points were considered to assess the effects of the treatments: pre-treatment (immediately before its introduction) and 72 h after its start. For each time point, the intensity of the symptoms was assessed on a numerical scale ranging from 0 to 10. The quality of life was assessed using the Palliative Outcome Scale (POS) [37,38], and the level of functionality was determined using the Palliative Performance Scale (PPS) [39]. We used the *International Classification of Primary Care, 2nd edition*, to classify clinical conditions and concomitant pathologies for which patients had been diagnosed, as recorded in their clinical files, to assess potential influences on the data collected regarding antibiotic use. We conducted stratified analyses based on all specific comorbidities to examine whether the effect of antibiotics differed across subgroups of patients with varying medical conditions.

The 72-h assessment timeframe was selected based on established clinical protocols for antibiotic therapy evaluation in palliative care settings. This period allows for adequate plasma steady-state concentrations to be achieved (typically after 5–6 doses for most antibiotics), enabling proper assessment of therapeutic response while avoiding missed opportunities for treatment optimization. Antibiotic steady-state pharmacokinetics demonstrate that therapeutic concentrations are achieved after 5–6 half-lives, allowing for adequate tissue penetration and bacterial suppression that correlates with symptom relief. This pharmacological principle supports the expectation of clinical response within 72 h rather than representing a placebo effect [40].

Standard re-evaluation protocols in community palliative care recommend assessment at 48–72 h to determine treatment continuation or modification. The decision to switch to a second-line antibiotic when initial treatment failure was determined followed established clinical protocols rather than confirmation bias, as delayed intervention could result in missed therapeutic opportunities and prolong patient suffering [41].

We collected data on all antibiotics used in our patient population. Information on these antibiotics (type, frequency of use) is included in the descriptive baseline characteristics of our study population. However, some antibiotics were excluded from the comparative statistical analysis that examined the effects of specific antibiotics on outcomes. This exclusion was necessary because, for each of the other individual antibiotics, we had a maximum of only two patients treated with that drug. With such extremely small sample sizes, any statistical comparisons would have been completely unreliable and uninterpretable. While we acknowledge that this decision could potentially introduce bias, we prioritized a robust statistical comparison of the four most commonly used antibiotics, as these represent the primary empirical treatment options in our setting. Future research with larger sample sizes could explore the effects of the less commonly used antibiotics in more detail.

We also want to highlight that this study was conducted within a CPCT, which means that no intravenous antibiotics were administered. All uses were oral, and in the few cases where we mentioned the need for alternative routes (in the cases of antibiotics with no statistical significance), intramuscular and subcutaneous routes were the most used. This happens because most of the patients had poor venous access and hypovolemic states that do not guarantee the stability of intravenous catheters for 24 h, which may compromise treatment, given that the team is only available from 8:00 a.m. to 8:00 p.m. on weekdays.

The effects of the four most prescribed drugs were compared: amoxicillin + clavulanic acid, cefixime, ciprofloxacin, and levofloxacin. The selection considered the most frequent infections in the population in question and the general recommendations for their treatment in the community [17,18,19,20]. The remaining antibiotics used had insufficient sample numbers for analysis.

We investigated differences between the effects of the mentioned antibiotics on survival, symptom intensity, functionality (measured through PPS), and quality of life (measured through POS).

For the purposes of this study, “complications” included any new infection, treatment failure, relapse, hospitalization, or adverse event requiring medical intervention.

We considered a treatment failure to have occurred when there were no major differences in symptoms (less than 3 on the numerical scale), and we considered a new infection to have occurred after an antibiotic course in the community (after 7 days of initiating the previous antibiotic treatment).

Respiratory secretions treated with antibiotics were all purulent pooled mucus, as predicted by Clark et al. (2009) [27], Sorenson (2000) [42], and van der Steen et al. (2002) [43], in studies dedicated to geriatric patients and patients with specialized palliative care needs. The goal of this approach is to reduce patient distress and improve comfort, while acknowledging that this practice is not universally recommended and should be considered on a case-by-case basis.

All patients were treated with antimuscarinics (ipratropium bromide in pressurized suspension, administered via spacer, at doses of 80 µg/day to 160 µg/day), systemic corticosteroids (dexamethasone, 4 to 8 mg/day), inhaled corticosteroids (fluticasone, 320 µg to 960 µg/day or budesonide, 400 µg to 1200 µg/day), and opioids (morphine equivalent dose of 40 to 320 mg/day). Some patients were on other medications, which may have influenced the results, but without statistical significance. To control for the potential confounding effects of these medications, we included them as covariates in our regression models. We used multiple regression and ANCOVA to adjust for the effects of other medications on symptom intensity, quality of life, and functionality.

Statistical analysis involved descriptive statistics (absolute and relative frequencies, means, and respective standard deviations) and inferential statistics. In this analysis, the Student’s t-test for independent samples, the Kruskal–Wallis test, the Chi-square test of independence, the Mann–Whitney test, the one-way ANOVA test, the Wilcoxon test for paired samples, and Spearman’s correlation coefficient were employed. The significance level for rejecting the null hypothesis was set at α ≤ 0.05.

To investigate differences in survival across the antibiotic groups (amoxicillin + clavulanic acid, cefixime, ciprofloxacin, and levofloxacin), we used the Chi-square test of independence. This test was appropriate because we were comparing the distribution of a categorical variable (survival status: time-to-event) across multiple independent groups (the antibiotic groups). The independent variable was the antibiotic group (categorical), and the dependent variable was survival (categorical). A *p*-value of <0.05 was considered statistically significant.

To assess the correlation between Palliative Outcome Scale (POS) scores and the intensity of symptoms, we used Spearman’s correlation coefficients. This non-parametric test was appropriate because we wanted to measure the strength and direction of the monotonic relationship between two continuous variables (POS score and symptom intensity), and we did not assume that the data were normally distributed. Both POS scores and symptom intensity were treated as continuous variables. A *p*-value of <0.05 was considered statistically significant.

To compare the intensity of symptoms before and after antibiotic treatment within each antibiotic group, we used the Wilcoxon signed-rank test for paired samples. This non-parametric test was appropriate because we were comparing two related samples (symptom intensity before and after treatment) within the same individuals, and we did not assume that the differences were normally distributed. The independent variable was treatment (before vs. after), and the dependent variable was symptom intensity (continuous). A *p*-value of <0.05 was considered statistically significant.

To compare the average age of medicated and non-medicated patients, we performed a Student’s t-test for independent samples. This test was appropriate because we were comparing the means of a continuous variable (age) between two independent groups (medicated with antibiotics vs. non-medicated). We assumed that the data were normally distributed. The independent variable was medicated or non-medicated (categorical), and the dependent variable was age (continuous). A *p*-value of <0.05 was considered statistically significant.

To compare the outcomes between three or more groups (Antibiotic A vs. Antibiotic B vs. Antibiotic C), we performed a one-way ANOVA test. This test was appropriate because we were comparing the means of a continuous variable between three or more independent groups. We assumed that the data were normally distributed. The independent variable was medicated with antibiotics or non-medicated (categorical), and the dependent variable was age (continuous). A *p*-value of <0.05 was considered statistically significant.

Statistical analysis was performed using SPSS (IBM Corp. Released 2021. IBM SPSS Statistics for Windows, Version 28.0. Armonk, NY, USA: IBM Corp).

## 5. Conclusions

The use of antibiotics to manage symptoms and enhance the quality of life for end-of-life patients remains a complex and nuanced issue in palliative care.

This retrospective study, conducted within a community palliative care team in Portugal, aimed to shed light on some important outcomes associated with antibiotic administration in highly clinically complex patients (oncological and non-oncological).

Our findings suggest that antibiotics, particularly second-line options such as ciprofloxacin, can offer significant relief from distressing symptoms and improve patients’ overall quality of life. However, this benefit must be carefully weighed against potential drawbacks.

While no significant impact on survival was observed, the heterogeneity of the patient sample and limitations in adverse effect data emphasize the need for individualized treatment approaches.

While antibiotics can provide comfort and relief, they are not without potential risks, including the development of antibiotic resistance and adverse events. A comprehensive assessment of each patient’s clinical condition, potential benefits, and potential harms is essential for responsible antibiotic prescription.

Future research should focus on refining diagnostic criteria, personalizing treatment strategies, and rigorously evaluating the long-term impact of antibiotic use on both individual patients and the broader community. By embracing evidence-based practices and prioritizing patient-centered care, we can optimize the use of antibiotics in palliative care, ensuring that these valuable medications are used effectively and responsibly to enhance the well-being of those nearing the end of life.

## Figures and Tables

**Table 1 antibiotics-14-00782-t001:** Intensity of symptoms before vs. 72 h after treatment.

	N (%)	Symptom Before Treatment	Symptom 72 h After Treatment	
		A ± SD	A ± SD	*p*
**Levofloxacin**	**15 (6.0)**	**7.75 ± 1.26**	**3.04 ± 1.98**	**0.001**
**Amoxicillin + Clavulanic Acid**	**10 (4.0)**	**7.35 ± 1.56**	**4.25 ± 2.57**	**0.001**
**Cefixime**	**5 (2.0)**	**7.17 ± 1.19**	**3.92 ± 2.61**	**0.003**
**Ciprofloxacin**	**13 (5.2)**	**6.78 ± 1.43**	**1.89 ± 1.64**	**0.001**
Other antibiotics (without statistical significance)	23 (9.2)	-	-	>0.05
**TOTAL with antibiotics**	**66 (26.5)**	**7.26 ± 1.44**	**3.85 ± 2.42**	**0.001**
**Without antibiotics**	**183 (73.5)**	**6.84 ± 2.13**	**4.52 ± 2.35**	**0.02**

A—average; SD—standard deviation; *p*—significance; bold—statistically significant differences; N—number of patients; %—percentage.

**Table 2 antibiotics-14-00782-t002:** POS before vs. 72 h after treatment.

	N (%)	POS Before Treatment	POS 72 h After Treatment	
		A ± SD	A ± SD	*p*
**Levofloxacin**	**15 (6.0)**	**3.21 ± 0.58**	**1.29 ± 0.69**	**0.001**
**Amoxicillin + Clavulanic Acid**	**10 (4.0)**	**3.15 ± 0.67**	**1.95 ± 1.05**	**0.001**
**Cefixime**	**5 (2.0)**	**3.42 ± 0.79**	**1.92 ± 1.31**	**0.010**
**Ciprofloxacin**	**13 (5.2)**	**2.89 ± 0.58**	**0.94 ± 0.80**	**0.001**
Other antibiotics (without statistical significance)	23 (9.2)	-	-	>0.05
**TOTAL with antibiotics**	**66 (26.5)**	**3.23 ± 0.69**	**1.89 ± 0.88**	**0.004**
**Without antibiotics**	**183 (73.5)**	**3.05 ± 0.84**	**1.94 ± 1.61**	**0.169**

A—average; SD—standard deviation; *p*—significance; bold—statistically significant differences; N—number of patients; %—percentage.

**Table 3 antibiotics-14-00782-t003:** PPS before vs. 72 h after treatment.

	N (%)	PPS Before Treatment	PPS 72 h After Treatment	
		A ± SD	A ± SD	*p*
Levofloxacin	**15 (6.0)**	15.00 ± 7.80	12.50 ± 6.75	0.058
Amoxicillin + Clavulanic Acid	**10 (4.0)**	15.00 ± 6.07	12.00 ± 5.23	0.124
**Cefixime**	**5 (2.0)**	**18.33** ± **8.34**	**12.50** ± **6.21**	**0.038**
**Ciprofloxacin**	**13 (5.2)**	**16.67 ± 6.86**	**12.22 ± 4.27**	**0.005**
Other antibiotics	23 (9.2)	-	-	>0.05
TOTAL with antibiotics	66 (26.5)	16.05 ± 7.01	13.25 ± 5.83	0.15
Without antibiotics	183 (73.5)	17.02 ± 6.12	13.10 ± 5.44	0.13

A—average; SD—standard deviation; *p*—significance; bold—statistically significant differences; N—number of patients; %—percentage.

## Data Availability

The data presented in this study are available on request from the corresponding author. The data are not publicly available due to privacy and ethical restrictions.

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
