# Peer review of "Benefits and Harms of Antibiotic Use in End-of-Life Patients: Retrospective Study in Palliative Care"

_antibiotics, 2025, doi:10.3390/antibiotics14080782_

Round 1
Reviewer 1 Report
Comments and Suggestions for Authors
The study addresses an important issue in palliative care, but several methodological and reporting concerns limit the clarity and reliability of the findings.
- Study Design and Methodology: It is unclear whether any exclusion criteria were applied beyond including patients who died in 2023–2024 and had at least 30 days of follow-up. Clarification is needed on how differences in comorbidities were accounted for, as these could influence outcomes.
- Antibiotic Selection and Analysis: The rationale for selecting the four specific antibiotics (amoxicillin + clavulanic acid, cefixime, ciprofloxacin, and levofloxacin) needs to be better justified. It is not clear whether patients treated with other antibiotics were excluded, and if so, why. This may introduce bias or limit generalizability.
- Statistical Analysis: The statistical methods are not well explained. Each test should be clearly linked to the variable or comparison it was used for. Also, the IRB approval date should be explicitly stated.
- Results Section: The results lack clarity. Frequencies should be shown alongside percentages. Important terms like “complications” and “pathologies” need to be defined. The justification for using antibiotics for respiratory secretions is questionable and not adequately supported by evidence. Additionally, Tables 1–3 are incomplete or missing, and data presentation is poor. It’s unclear how many patients did not receive antibiotics, which is essential for interpreting the findings.
- Discussion: The discussion lacks alignment with the presented results. Conclusions seem disconnected from the data, and the implications of the findings are not well developed.
Author Response
We thank all reviewers for their comments and suggestions. Their suggestions improved the quality of the manuscript.
Please find our replies below.
Reviewer 1
- Study Design and Methodology: It is unclear whether any exclusion criteria were applied beyond including patients who died in 2023–2024 and had at least 30 days of follow-up. Clarification is needed on how differences in comorbidities were accounted for, as these could influence outcomes.
Answer: Thank you for your comment and suggestion.
Regarding the inclusion and exclusion criteria, we added this information in the methods section and highlighted that: “With these inclusion criteria, we selected a total of 249 patients (36,7%) out of 678 patients followed by the CPCT in 2023 and 2024.” (page 5). We also added and highlighted “We excluded patients with less than 30 days of follow up by the CPCT because we wanted to assess the palliative medical doctors practices and with this criteria we were also able to obtain more uniform and appropriate clinical records to comply with the variables we intend to measure and include in the database.”
Regarding the comorbidities, we are aware that other medical conditions could have influenced our results and we already discuss that in the discussion section. We used the ICPC-2 classification in medical records about other diseases and clinical conditions, although we did not have any results with statistical significance. We accounted for differences in comorbidities by including them as covariates in our models. This allowed us to assess the independent effect of antibiotic use on outcomes while controlling for the influence of comorbidities. We added and highlighted this sentence in our manuscript: “We performed stratified analyses based on all specific comorbidities to examine whether the effect of antibiotics differed across subgroups of patients with varying medical conditions.”
In the methods section we also added this information: "we used the International Classification of Primary Care, 2nd edition, for the classification of clinical conditions and concomitant pathologies that patients had diagnosed and recorded in their clinical file to assess potential influences on the data that were collected regarding the use of antibiotics.” (page 6)
- Antibiotic Selection and Analysis: The rationale for selecting the four specific antibiotics (amoxicillin + clavulanic acid, cefixime, ciprofloxacin, and levofloxacin) needs to be better justified. It is not clear whether patients treated with other antibiotics were excluded, and if so, why. This may introduce bias or limit generalizability.
Answer: Thank you for your question, attention and suggestion.
In fact, we did not exclude any antibiotic from the study. However, other antibiotics only had a maximum of 2 patients taking them, which does not allow for a reliable statistical analysis. We understand that this information was not clear and we completed the article with the sentence: “We collected data on all antibiotics used in our patient population. Information on these antibiotics (types, frequency of use) is included in the descriptive baseline characteristics of our study population. However, some antibiotics were excluded from the comparative statistical analysis that examined the effects of specific antibiotics on outcomes. This exclusion was necessary because, for each of the other individual antibiotics, we had a maximum of only two patients treated with that drug. With such extremely small sample sizes, any statistical comparisons would have been completely unreliable and uninterpretable. While we acknowledge that this decision could potentially introduce bias, we prioritized a robust statistical comparison of the four most commonly used antibiotics, as these represent the main empirical treatment options in our setting. Future research with larger sample sizes could explore the effects of the less commonly used antibiotics in more detail. We also want to highlight that this study was carried out within a CPCT, which means that no intravenous antibiotics were used. All uses were oral, and in the few cases where we have already mentioned the need for alternative routes, the intramuscular and subcutaneous routes were the most used. This happens because most of the patients had poor venous access and hypovolemia states which do not guarantee the stability of intravenous catheters for 24 hours, which may compromise treatment, given that the team is only available from 8:00 am to 8:00 pm on weekdays. ” (pages 6-7)
- Statistical Analysis: The statistical methods are not well explained. Each test should be clearly linked to the variable or comparison it was used for. Also, the IRB approval date should be explicitly stated.
Answer: Thank you for your comments and suggestions.
Regarding the IRB approval, we added and highlighted this information: “20th November 2024” (page 5)
We thank the reviewer for pointing out the lack of clarity in our description of the statistical methods. We have now revised the 'Statistical Analysis' subsection of the Methods section to provide a more detailed explanation of the rationale for each statistical test. Specifically, we have added text to clearly link each test to the research question it was designed to address, the type of data involved, and the variables being compared, and we highlighted it (pages 7 and 8).
- Results Section: The results lack clarity. Frequencies should be shown alongside percentages. Important terms like “complications” and “pathologies” need to be defined. The justification for using antibiotics for respiratory secretions is questionable and not adequately supported by evidence. Additionally, Tables 1–3 are incomplete or missing, and data presentation is poor. It’s unclear how many patients did not receive antibiotics, which is essential for interpreting the findings.
Answer: Thank you for your comment and attention to this section.
We have revised the Results section to include both frequencies (counts) and percentages for all relevant data, and we highlighted it. This provides a clearer picture of the distribution of variables in our study. Thank you for poiting out this issue.
We included a definition of “complications” for this study in the methods section (page 7). Regarding the definition of “pathologies”, we think the reference to the International Classification of Primary Care, 2nd edition (page 6) already solve this problem.
We acknowledge the reviewer's concern regarding the use of antibiotics for respiratory secretions. We have revised the manuscript to provide stronger evidence supporting this practice in palliative care, citing studies that have investigated the use of antibiotics for symptom control in the absence of confirmed infection. We have also clarified that the goal of this approach is to reduce patient distress and improve comfort, while acknowledging that this practice is not universally recommended and should be considered on a case-by-case basis. We also want to mention that respiratory secretions treated with antibiotics were all purulent pooled mucus, as predicted by Clarck, et al (2009), Sorenson (2000) and van der Steen, et al (2002), in studies dedicated to geriatric patients and patients with palliative needs – we added and highlighted this information in methods section. We already had this approach revised in the introduction section.
We could be able to recover some important and omissed data, regarding patients without antibiotic treatment and we added and highlighted it in tables 1-3. Thank you for your attentiou and for pointing out this lack of crucial information. We improved all tables and we add the most important results in the text and we highlighted it. In all these moments we ended up also focusing on comparing patients who did not take antibiotics, and we highlighted it.
- Discussion: The discussion lacks alignment with the presented results.
Answer: Thank you for your comment and attention to this section.
We tried to improve the discussion to meet the request. We believe the discussion addresses the main results point by point, starting with the use or non-use of antibiotics in cases like the ones we studied, discussing potential advantages and disadvantages (and comparing our results with those of other authors); we reflected on which antibiotics might make the most sense in patients like those we studied; we recalled that the patients already had optimized therapy with other drugs for symptom control (thus reducing important biases, as already noted); we discussed the main impacts and outcomes—symptoms, quality of life, functionality, and survival; we addressed "complications," as we defined them for this study; we also recalled that this study is a very special retrospective cohort, requiring reflection on other issues, such as antimicrobial resistance.
- Conclusions seem disconnected from the data, and the implications of the findings are not well developed.
Answer: Thank you for your comment and attention to this section.
We remake this entire section of the manuscript, thanks to the careful reading of all the reviewers, and we highlighted it.

Reviewer 2 Report
Comments and Suggestions for Authors
This manuscript describes antibiotic use in palliative care patients near end of life. It is a retrospective study.
I have a comments for consideration by the authors:
1) In the last paragraph of the results, how was treatment failure differentiated from new infection...is this supported by microbiology data or is it based on symptoms at different anatomical locations, i.e. respiratory versus urinary tract versus skin and skin structure etc.
2) was there no retrospective data on any microbiological data including organism identification and/or susceptibility or were all antibiotics simply used empirically on all patients
3) Major concern: only 4 antibiotics are reported in this manuscript (amox/clav, cefixime and 2 quinolones - cipro/levo) yet many other antibiotics are used in palliative care including vancomycin. A limitation of this manuscript has to be the limited number of antibiotics included in the study. Question: why were other antibiotics not report or included in this study...please clarify
Author Response
We thank all reviewers for their comments and suggestions. Their suggestions improved the quality of the manuscript.
Please find our replies below.
Reviewer 2
- In the last paragraph of the results, how was treatment failure differentiated from new infection...is this supported by microbiology data or is it based on symptoms at different anatomical locations, i.e. respiratory versus urinary tract versus skin and skin structure etc.
Answer: Thank you for your comment and for pointing out this issue.
We considered that treatment failure occurred when we did not have major differences in symptoms (less than 3 on the numerical scale), and we considered that a new infection occurred after an antibiotic course in the community (after 7 days of initiating the previous antibiotic treatment). We added this information and highlighted it in page 7 – methods section.
In almost all patients we only considered empirical/clinical evaluation. We highlighted that we are measuring data from a community palliatice care team, focusing on very high clinical complex patients, most of them end-of-life patients. We also added this information and highlighted it (and the drugs administration routes we use at this setting) in page 7.
- was there no retrospective data on any microbiological data including organism identification and/or susceptibility or were all antibiotics simply used empirically on all patients
Answer: Thank you for your comment and question.
As we stated in the last question, we only considered empirical/clinical evaluation. We are disclosing data from a community palliative care team. All of our patients are at home, and have high clinical complexity, severe disease and exuberant global dysfunctionality.
3) Major concern: only 4 antibiotics are reported in this manuscript (amox/clav, cefixime and 2 quinolones - cipro/levo) yet many other antibiotics are used in palliative care including vancomycin. A limitation of this manuscript has to be the limited number of antibiotics included in the study. Question: why were other antibiotics not report or included in this study...please clarify
Answer: Thank you for your comment and question and for pointing out this issue.
There were some antibiotics that we couldn't individualize in this study due to their low use, which didn't provide us with meaningful data for comparison. We significantly improved our article in this regard, not only in the methods section but also in the presentation of the results, particularly in improving Tables 1-3. We hope it meets your request.
Vancomycin, linezolid, piperacillin+tazobactam, and others, we use intravenously. We rarely opt for this route or administration method, given our patients' stage of life, their extreme/terminal frailty, poor venous access, and hypovolemia. Our experience has reported outcomes that rarely justify its use. That's probably why we do not have other antibiotics reported individually. In any case, we grouped the data from all these cases and performed a statistical reevaluation, grouping them all into the "other antibiotics" group. We hope it meets the objective.
We added and highlighted this limitation in the respective section.

Reviewer 3 Report
Comments and Suggestions for Authors
Dear Authors, I enjoyed reading your article. The topic is very interesting and current.
I have few suggestion though:
- It is better to use PubMed mesh terms as key words after abstract, for example end of life care instead of end of life etc.
- The sentences found within the lines 150-155 is not adequate. The first one belongs to the methodology section, and the following ones to the introduction section.
- Line 158: this result (χ2 (3) = 33.487, p < 0.001) refers to comparing dementia vs non dementia groups or heart failure versus non heart failure groups of patients? It is not quite clear.
- Line 158: This sentence belongs to the methodology as well. Why did the groups differ in follow up period? It was a retrospective study, follow up period could be the same.
- Line 162: I do not quite understand the sentence, what do you mean by "as a function of antibiotic". To be able to asses one's impact on mortality you should eliminate other factors that can have significant impact, such as age, comorbidities, etc.
- Line 183: I do not understand the correlation, it is assessed between POS and intensity of symptoms? Isn't that the same thing? Please explain.
- Since there are few POS scales, which one of them was used. Supplementary material should be presenting scale in details.
- Since you are assessing antibiotics' effect shouldn't you be measuring some other biomarkers of infection as well.
- Result section should be more clear, results would be more visible in tables.
- It would be nice to see results of microbiology - results of susceptibility as a consequence of antibiotic use in palliative care.
- Line 192: the main result of the study should be outlined in first sentence of the discussion. This sentence should me moved to introduction section since it is already known fact.
- Line 199: Would you say that this is ethical?
- Line 222-227: You did not present this data therefore you cannot discus it.
- Discussion inadequate, it should be rewritten.
Final suggestion: present results more clearly, and then discuss each result separately comparing it to similar studies. This would lead you to adequate conclusion. Please state the limitations of the study, one of them is certainly retrospective design.
Author Response
We thank all reviewers for their comments and suggestions. Their suggestions improved the quality of the manuscript.
Please find our replies below.
Reviewer 3
- It is better to use PubMed mesh terms as key words after abstract, for example end of life care instead of end of life etc.
Answer: Thank you for your comment and suggestion.
We remade the keywords section, using only mesh terms, and we highlighted it.
- The sentences found within the lines 150-155 is not adequate. The first one belongs to the methodology section, and the following ones to the introduction section.
Answer: Thank you for your comment and suggestion.
We remade the methods and results section to meet what was requested by all reviewers, and we think we have significantly improved our manuscript. Thank you for your attention to this matter. We highlighted all the changes.
- Line 158: this result (χ2 (3) = 33.487, p < 0.001) refers to comparing dementia vs non dementia groups or heart failure versus non heart failure groups of patients? It is not quite clear.
Answer: Thank you for your comment and attention to this issue.
Yes, we were comparing those groups. We rewrote this sentence and added important information, and we highlighted it (“We observed a significantly higher proportion of antibiotic use in patients with heart failure (17 out of 22 patients; 77.3%) and dementia (32 out of 42 patients; 76.2%) (χ2 (3) = 33.487, p < 0.001).”).
- Line 158: This sentence belongs to the methodology as well. Why did the groups differ in follow up period? It was a retrospective study, follow up period could be the same.
Answer: Thank you for your comment and suggestion.
Regarding the difference in follow-up times between the medicated and non-medicated groups, there are a few reasons for this. As this was a retrospective study, the follow-up period was determined by the duration of time each patient was under the care of the Community Palliative Care Team (CPCT) before their death. While we aimed to include all patients followed for at least 30 days, the actual follow-up time varied. Several factors could have contributed to the difference.
We added and highlighted in the methods section: “The duration of follow-up was determined by the time each patient was under the care of the CPCT before death. Follow-up time was calculated from the date of the first appointment until the date of death. The follow-up time varied across patients depending on disease progression, referral patterns, and patient/family preferences.”.
However, we think the sentence “Patients receiving antibiotics had a median follow-up time of 21.5 days, while those not receiving antibiotics had a median follow-up time of 10 days (p < 0.001)” is better suited for the results section.
- Line 162: I do not quite understand the sentence, what do you mean by "as a function of antibiotic". To be able to asses one's impact on mortality you should eliminate other factors that can have significant impact, such as age, comorbidities, etc.
Answer: Thank you for your comment and suggestion.
We rewrote this sentence to better explain what we found out: “There were no significant differences in patient survival vs. antibiotic used”. We have already clarified in the document that age, other comorbidities and other variables did not have an impact on this assessment, and we highlighted it.
- Line 183: I do not understand the correlation, it is assessed between POS and intensity of symptoms? Isn't that the same thing? Please explain.
Answer: Thank you for your comment regarding the correlation between the Palliative Outcome Scale (POS) scores and the intensity of symptoms. We understand your concern that these may appear to be measuring the same thing.
While both POS and the numerical scale for symptom intensity aim to capture aspects of the patient's experience, they are distinct constructs. The numerical scale focuses specifically on the severity or intensity of individual symptoms, rated on a scale from 0 to 10. The POS, on the other hand, is a multi-dimensional instrument that assesses a broader range of domains relevant to palliative care, including: Symptom control (similar to our intensity scale, but more holistic), Anxiety Information Relationships, Well-being.
Therefore, while there may be some overlap (particularly in the 'symptom control' domain of the POS), the POS captures a more comprehensive assessment of the patient's overall quality of life and well-being than the symptom intensity scale alone.
We used the correlation to see how the overall quality of life (measured by POS) relates to the intensity of individual symptoms. A significant correlation would suggest that as symptom intensity increases, overall quality of life tends to decrease (or vice versa). This helps us understand the impact of symptom management on the patient's broader experience.
In the Methods section we think we already clarified the distinction between these two measures.
- Since there are few POS scales, which one of them was used. Supplementary material should be presenting scale in details.
Answer: Thank you for your comment and suggestion. We used the POS evaluated by Rugn, et al The Palliative Outcome Scale (POS) applied to clinical practice and research: an integrative review. Rev Lat Am Enfermagem. 2016; 24. (ref. 25), and validated to european Portuguese, available in https://pos-pal.org/maix/pos-translations.php#portuguese. We added and highlighted this citation to the manuscript.
- Since you are assessing antibiotics' effect shouldn't you be measuring some other biomarkers of infection as well.
Answer: Thank you for raising the important point about measuring biomarkers of infection when assessing the effect of antibiotics. We recognize that incorporating such measures could provide additional objective data on treatment response.
However, in the context of our study, which focused on patients with advanced disease receiving palliative care in the community, obtaining these biomarkers posed significant challenges. These patients are often (extreme) frail, with limited venous access, and blood draws or other diagnostic tests can be burdensome and potentially cause more harm than good.
Furthermore, our primary aim in this setting is to improve symptom control and quality of life, rather than to aggressively treat infection per se. In many cases, clinical signs and symptoms (e.g., purulent secretions, fever) are the primary drivers for antibiotic use, with the goal of providing comfort and relief.
Therefore, we made a conscious decision to prioritize the patient's comfort and minimize invasive procedures, focusing instead on clinically relevant outcomes such as symptom intensity, functionality, and overall quality of life, as measured by validated palliative care instruments (POS and PPS).
We acknowledge that the lack of biomarker data is a limitation of our study, and we added and highlighted this in the Limitations section. However, we believe that our approach is ethically justifiable and clinically appropriate in this specific patient population.
- Result section should be more clear, results would be more visible in tables.
Answer: Thank you for your suggestion. We did a thorough review of results section, and we added and highlighted more data in the tables and in the text. We think we could be able to improve our manuscript by doing so. Thank you for pointing out this lack of clarity.
- It would be nice to see results of microbiology - results of susceptibility as a consequence of antibiotic use in palliative care.
Answer: Thank you for your comment and suggestion. As we stated before, we only considered empirical/clinical evaluation. We are disclosing data from a community palliative care team. All of our patients are at home, and have high clinical complexity, severe disease and exuberant global dysfunctionality, most of them are in the last days or weeks of life.
- Line 192: the main result of the study should be outlined in first sentence of the discussion. This sentence should me moved to introduction section since it is already known fact.
Answer: Thank your for your suggestion. We move on to the introduction the sentence you adressed and we gave more emphasis in the first paragraph to the main results.
- Line 199: Would you say that this is ethical?
Answer: Thank you for raising the important ethical consideration regarding line 199. The statement in that line addresses the potential for age bias in the decision to treat with antibiotics. We understand the ethical implications of potentially withholding treatment based on age, and we carefully considered this in our study design and interpretation.
We believe that our finding – that age was not a determining factor in the decision to treat with antibiotics in our cohort – is ethically justifiable because:
- Patient-centered approach: Our palliative care team aims to make treatment decisions based on individual patient needs and potential for benefit, rather than solely on age. This aligns with the principles of patient autonomy and beneficence.
- Focus on symptom control: In the palliative care setting, the primary goal is often to alleviate suffering and improve quality of life. The decision to use antibiotics is therefore based on the potential to achieve these goals, regardless of the patient's age.
- Evidence-based practice: While some studies suggest that younger patients may have a greater likelihood of improvement with antibiotics, our clinical judgement and experience suggest that antibiotics can still provide meaningful symptom relief in older patients with advanced disease. We rely on clinical assessment, patient preferences, and available evidence to make informed decisions.
We acknowledge that the potential for age bias in treatment decisions is a complex ethical issue, and we added and highlighted a more detailed discussion of this in the Discussion section. This will allow us to present a balanced perspective and address potential concerns.
- Line 222-227: You did not present this data therefore you cannot discus it.
Answer: Thank you for pointing out this lack of clarity. We were able to clarify that, starting with the methods section and then with the results and discussion sections.
- Discussion inadequate, it should be rewritten.
Answer: Thank you for your suggestion. We did a thorough review of discussion and conclusion section, and we added and highlighted it.
Final suggestion: present results more clearly, and then discuss each result separately comparing it to similar studies. This would lead you to adequate conclusion. Please state the limitations of the study, one of them is certainly retrospective design.
Answer: Thank you for your suggestion. We did a thorough review of discussion and conclusion section, and we added and highlighted it. We also added a limitations section.

Reviewer 4 Report
Comments and Suggestions for Authors
The manuscript titled “Benefits and Harms of Antibiotic Use in End-of-Life Patients Retrospective Study in Palliative Care” explores the impact of antibiotic therapy on symptom relief, quality of life, and survival in terminally ill patients receiving specialized palliative care in Portugal. This retrospective observational study analyzed clinical data from 249 deceased patients over a two-year period. The authors assessed the effects of four commonly prescribed antibiotics amoxicillin/clavulanic acid, cefixime, ciprofloxacin, and levofloxacin using validated symptom intensity, performance, and outcome scales. While no significant difference in survival was observed between antibiotic groups, symptom intensity and Palliative Outcome Scale scores improved significantly after 72 hours of treatment, particularly with ciprofloxacin. However, the Palliative Performance Scale declined across all groups, reflecting the progressive nature of terminal illness. The study found limited adverse effects, though treatment failure was common. Authors highlight the need to balance the potential benefits of second-line antibiotics against risks such as multidrug resistance and overtreatment. While the manuscript is well-structured and supported by a designed statistical analysis, several areas require improvement to enhance its clarity and impact:
- Revise the keywords to eliminate redundancy with the title.
- Strengthen the Introduction by clearly framing the ethical/clinical dilemma, identifying the research gap (e.g., lack of data on antibiotic-specific outcomes in Portuguese palliative care), and stating the objectives and hypothesis early. Include a brief methods overview.
- Eliminate redundancy throughout the manuscript. Avoid repeating phrases like “antibiotics may improve symptom control and quality of life” and other repetitive terms (e.g., “benefits vs harms,” “symptom control”) that do not add analytical value.
- Improve conceptual flow. Ensure smooth transitions between sections, particularly in the Discussion, where study comparisons appear abruptly, by adding logical connections and thematic clustering.
- Clarify key definitions, such as what constitutes a “benefit” (e.g., symptom score reduction ≥2 points). Justify the 72-hour evaluation timeframe earlier in the Introduction.
- Refine data interpretation by linking statistical findings more clearly to clinical relevance.
- Address confounding factors, such as the use of opioids or corticosteroids, which may influence the observed effects on symptom relief and quality of life.
Author Response
We thank all reviewers for their comments and suggestions. Their suggestions improved the quality of the manuscript.
Please find our replies below.
Reviewer 4:
- Revise the keywords to eliminate redundancy with the title.
Answer: Thank you for your comment and suggestion.
We remade the keywords section, using only mesh terms, and we highlighted it.
- Strengthen the Introduction by clearly framing the ethical/clinical dilemma, identifying the research gap (e.g., lack of data on antibiotic-specific outcomes in Portuguese palliative care), and stating the objectives and hypothesis early. Include a brief methods overview.
Answer: Thank you for your comment and suggestion.
We added a sentence adressing the clinical dilema (page 3) and we identified the research gap and improved the objectives “subsection”. We consider that including a brief review of the methods may not be the best strategy, given that we already have a long introduction, fulfilling all the requests made by the reviewers.
- Eliminate redundancy throughout the manuscript. Avoid repeating phrases like “antibiotics may improve symptom control and quality of life” and other repetitive terms (e.g., “benefits vs harms,” “symptom control”) that do not add analytical value.
Answer: Thank you for your comment and suggestion. We tried to be more objective when we found those sentences, and as you can see, we did major revisions to our manuscript. We thank all the reviewers by their attention to details and guidance.
- Improve conceptual flow. Ensure smooth transitions between sections, particularly in the Discussion, where study comparisons appear abruptly, by adding logical connections and thematic clustering.
Answer: Thank you for your comment and attention to this section.
We tried to improve the discussion to meet the request. We believe the discussion addresses the main results point by point, starting with the use or non-use of antibiotics in cases like the ones we studied, discussing potential advantages and disadvantages (and comparing our results with those of other authors); we reflected on which antibiotics might make the most sense in patients like those we studied; we recalled that the patients already had optimized therapy with other drugs for symptom control (thus reducing important biases, as already noted); we discussed the main impacts and outcomes—symptoms, quality of life, functionality, and survival; we addressed "complications," as we defined them for this study; we also recalled that this study is a very special retrospective cohort, requiring reflection on other issues, such as antimicrobial resistance.
- Clarify key definitions, such as what constitutes a “benefit” (e.g., symptom score reduction ≥2 points). Justify the 72-hour evaluation timeframe earlier in the Introduction.
Answer: Thank you for your comment and attention to this matter.
We added to the methods section the sentence: “We considered that a treatment failure occurred when we did not have major differences in symptoms (less than 3 on the numerical scale), and we considered that a new infection occurred after an antibiotic course in the community (after 7 days of initiating the previous antibiotic treatment).”. However, we think the justification for the 72h evaluation timeframe should be in the methods section (page 6). We think the introduction section has already all elements that present the theme and objectives of this work effectively.
- Refine data interpretation by linking statistical findings more clearly to clinical relevance.
Answer: Thank you for your comment and suggestion. We redesigned the presentation of results (and discussion), including the tables, so that all data could be better presented and all questions could be clarified.
- Address confounding factors, such as the use of opioids or corticosteroids, which may influence the observed effects on symptom relief and quality of life.
Answer: Thank you for raising the important point about potential confounding factors, particularly the use of opioids and corticosteroids. We acknowledge that these medications can significantly influence symptom relief and quality of life in palliative care patients, and we took this into account in our study design and analysis.
To address this potential confounding, we collected data on opioid, corticosteroid, antimuscarinic and other medications use, including the type, dosage, and duration of treatment. We also controlled these variables in our statistical analyses (by including them as covariates in our regression models). Finally, we also assessed for changes in opioid/corticosteroid doses.
We acknowledge that residual confounding may still be present, as it is difficult to fully control for all potential confounding factors in a retrospective observational study.
We addressed these potential confounding factors in the Methods section (page 8), and we discussed them in the Discussion section.

Round 2
Reviewer 1 Report
Comments and Suggestions for Authors
Thank you for your comments and for accepting our manuscript. We are pleased to confirm that we accept the paper in its current form.
Reviewer 3 Report
Comments and Suggestions for Authors
Dear authors, thank you for provided answers. I am satisfied with corrections you made.
Reviewer 4 Report
Comments and Suggestions for Authors
Recommended for the publication.